# Mutational signatures reveal the dynamic interplay of risk factors and cellular processes during liver tumorigenesis

Eric Letouzé [1,2,3,4], Jayendra Shinde [1,2,3,4], Victor Renault [5], Gabrielle Couchy[1,2,3,4], Jean-Frédéric Blanc[6,7], Emmanuel Tubacher[5], Quentin Bayard[1,2,3,4], Delphine Bacq[8], Vincent Meyer[8], Jérémy Semhoun[5], Paulette Bioulac-Sage[6,9], Sophie Prévôt[10], Daniel Azoulay[11,12], Valérie Paradis[13], Sandrine Imbeaud [1,2,3,4], Jean-François Deleuze[8] & Jessica Zucman-Rossi [1,2,3,4,14]

Genomic alterations driving tumorigenesis result from the interaction of environmental exposures and endogenous cellular processes. With a diversity of risk factors, liver cancer is an ideal model to study these interactions. Here, we analyze the whole genomes of 44 new and 264 published liver cancers and we identify 10 mutational and 6 structural rearrangement signatures showing distinct relationships with environmental exposures, replication, transcription, and driver genes. The liver cancer-specific signature 16, associated with alcohol, displays a unique feature of transcription-coupled damage and is the main source of *CTNNB1* mutations. Flood of insertions/deletions (indels) are identified in very highly expressed hepato-specific genes, likely resulting from replication-transcription collisions. Reconstruction of sub-clonal architecture reveals mutational signature evolution during tumor development exemplified by the vanishing of aflatoxin B1 signature in African migrants. Finally, chromosome duplications occur late and may represent rate-limiting events in tumorigenesis. These findings shed new light on the natural history of liver cancers.

[1] INSERM, UMR-1162, Génomique Fonctionnelle des Tumeurs Solides, Equipe Labellisée Ligue Contre le Cancer, Institut Universitaire d'Hématologie, Paris, 75010, France. [2] Université Paris Descartes, Labex Immuno-Oncology, Sorbonne Paris Cité, Faculté de Médecine, Paris, 75006, France. [3] Université Paris 13, Sorbonne Paris Cité, Unité de Formation et de Recherche Santé, Médecine, Biologie Humaine, Bobigny, 93000, France. [4] Université Paris Diderot, Paris, 75013, France. [5] Laboratory for Bioinformatics, Fondation Jean Dausset—CEPH, Paris, 75010, France. [6] Université Bordeaux, Bordeaux Research in Translational Oncology, Bordeaux, 33000, France. [7] Service Hépato-Gastroentérologie et Oncologie Digestive, Centre Medico-Chirurgical Magellan, Hôpital Haut-Lévêque, Centre Hospitalier Universitaire de Bordeaux, Bordeaux, 33000, France. [8] Centre National de Recherche en Génomique Humaine, CEA, Evry, 91000, France. [9] Service de Pathologie, Hôpital Pellegrin, Centre Hospitalier Universitaire de Bordeaux, Bordeaux, 33000, France. [10] AP-HP, Hôpital Antoine-Béclère, Service d'anatomie pathologique, Clamart, 92140, France. [11] Université Paris Est Créteil, Créteil, 94010, France. [12] AP-HP, Groupe Hospitalier Henri Mondor, Département de Chirurgie Hépato-Biliaire et Transplantation Hépatique, Créteil, 94010, France. [13] Service d'Anatomopathologie, Hôpital Beaujon, Clichy, 92110, France. [14] Assistance Publique-Hôpitaux de Paris, Hopital Europeen Georges Pompidou, 75015 Paris, France. Eric Letouzé and Jayendra Shinde contributed equally to this work. Correspondence and requests for materials should be addressed to E.L. (email: eric.letouze@inserm.fr) or to J.Z.-R. (email: jessica.zucman-rossi@inserm.fr)

Somatic mutations in human cancers result from diverse processes including replication errors, spontaneous or enzymatic conversions and exposure to endogenous or exogenous DNA damaging agents[1]. Each of these processes leaves a characteristic pattern of mutations on the tumor genome or mutational signature. Mathematical extraction of mutational signatures[2] in large pan-cancer series revealed more than 20 different signatures[3]. Several signatures could be associated with known (smoking, UV light) or new (APOBEC mutagenesis)

etiologies, but half of them remain unexplained. In addition, mutations do not occur uniformly over the genome. Local mutation rates are modulated by cellular processes like replication, transcription[4] and chromatin organization[5]. General trends have been identified, like the higher mutation rate in highly expressed and late-replicating regions[4]. However, recent studies have shown that different mutational processes can have different types of interactions with these genomic features[6, 7]. Unraveling the etiology of mutational signatures and their interactions with

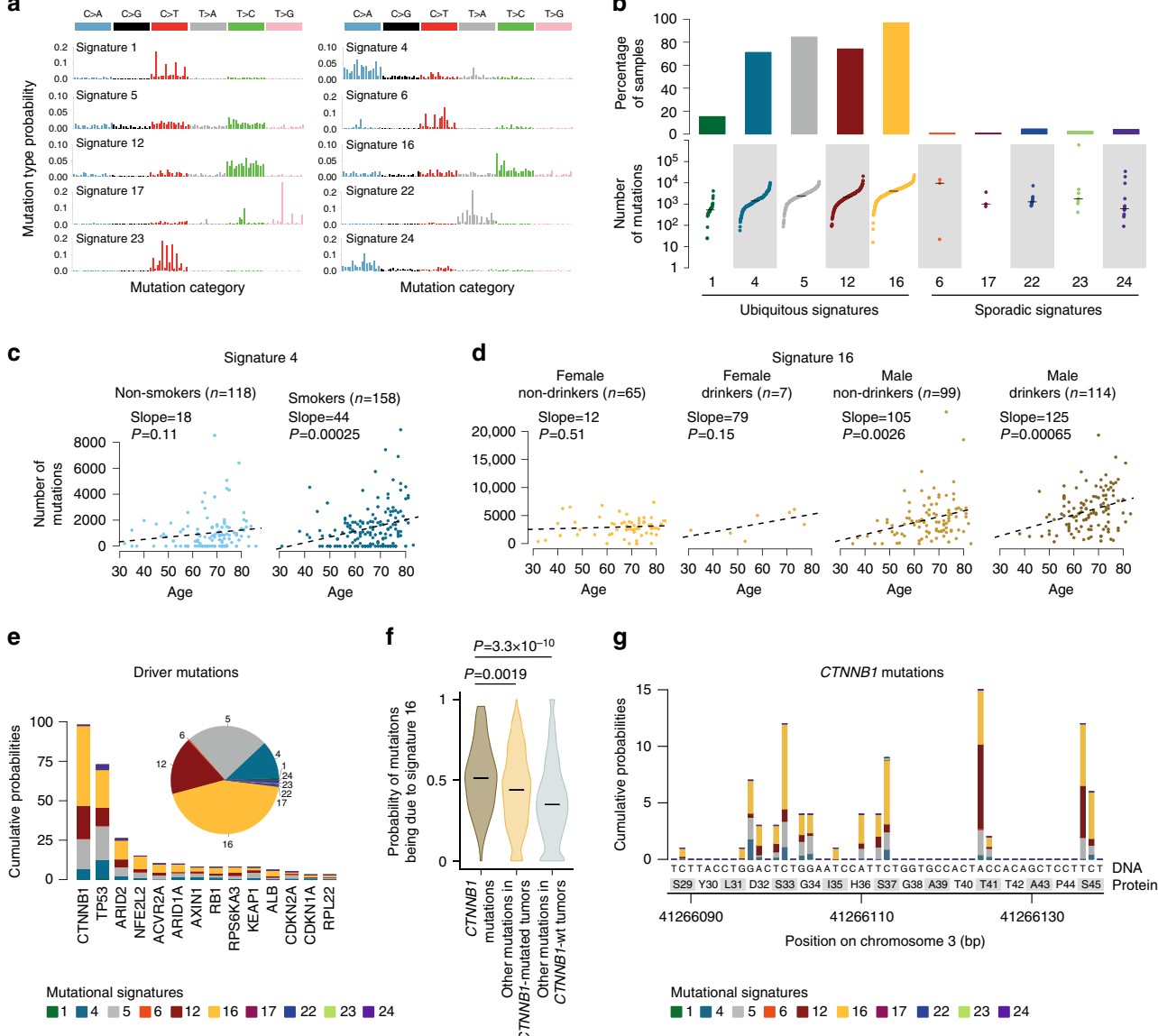

**Fig. 1** Mutational processes are modulated by risk factors and preferentially alter specific hotspots. **a** Summary of the 10 signatures (COSMIC nomenclature) known to be operative in liver cancers analyzed in our combined series of 299 HCC. Each signature is displayed according to the 96-substitution classification defined by the substitution type and sequence context immediately 5′ and 3′ to the mutated base. **b** Proportion of samples in which each signature was detected (top) and distribution of mutation counts for each signature in the relevant samples (bottom). **c**, **d** Correlation of mutational signatures with risk factors and age. The number of mutations attributed to signatures 4 **c** and 16 **d** is represented as a function of age for patient groups stratified according to the risk factors significantly associated with each signature (Supplementary Table 5). **e** Distribution of mutational signatures associated with driver gene mutations. We estimated the probability of each driver gene mutation being due to each mutational process. We then summed these probabilities over all mutations and signatures to obtain the cumulative probabilities across all driver gene mutations (pie chart) and for each driver gene separately (barplot). **f** *CTNNB1* mutations (left) overall have higher probabilities being due to signature 16 than other mutations in the same samples (middle) and in other samples (right). The violin plots represent the distribution of probabilities for each group of mutations and horizontal segments highlight median values. **g** Distribution of mutational signatures associated with *CTNNB1* mutations across the main oncogenic hotspots. The probability of each *CTNNB1* mutation being due to each mutational process was estimated and summed across each position between amino acids S29 and S45, the main oncogenic hotspots of *CTNNB1*. The DNA and protein sequence are depicted below the barplot

cellular processes thus remain key questions to better understand cancer development.

Hepatocellular carcinoma (HCC) predominantly develops in male and is related to various etiologies including viral infection (HBV, HCV, AAV2[8]), alcohol abuse or metabolic syndrome[9]. In addition, alimentary exposures to aflatoxin B1 and aristolochic acid carcinogens promote liver tumorigenesis related to specific mutational signatures[10, 11]. Also, whole exome and whole genome sequencing studies by us[11, 12] and others[13–15] revealed tens of driver genes recurrently altered in HCC, some of which being preferentially associated with specific risk factors. For example, *CTNNB1* are more frequent in alcohol-related cases, whereas *TP53* mutations are more frequent in HBV-related cases[11]. However, the molecular mechanisms giving rise to these mutations and their interaction with risk factors remain incompletely understood.

Here, we report the comprehensive analysis of mutational signatures in >300 liver tumor genomes. We identify 10 mutational signatures, including ubiquitous signatures modulated by risk factors like alcohol or tobacco, and sporadic signatures restricted to specific etiologies. Each signature is modulated differently by DNA replication and transcription depending on the underlying mutational process, and has a different propensity to target specific driver genes. Finally, we unravel the clonal architecture of tumors and we reconstruct the temporal evolution of driver mutations and signatures in each tumor.

## Results

**Whole genome sequencing of 44 liver tumors**. To explore the diversity of genomic alteration signatures in HCC, we sequenced the genomes of 44 liver tumors surgically resected in Europe (Supplementary Table 1). These included 35 HCC, mostly developed in absence of cirrhosis (25 non-fibrotic, 7 chronic hepatitis and 3 cirrhotic livers) and associated with diverse etiological backgrounds including HBV ($n = 5$) or HCV infection ($n = 4$), alcohol ($n = 12$), metabolic syndrome ($n = 7$), hemochromatosis ($n = 1$), and without etiology ($n = 6$). Four fibrolamellar carcinomas (FLC) and five hepatocellular adenomas (HCA) were included for comparison. The average depth of whole genome sequencing was 92-fold for tumors, allowing clonal architecture analysis, and 68-fold for matched non-tumor liver samples (Supplementary Table 2). Consistent with previous reports, we identified a median of 13,740 clonal single-nucleotide variants (SNVs, 4.9 per Mb), 756 small insertions and deletions (indels) and 46 structural variants per tumor, as well as 12 HBV insertion sites (Supplementary Data 1, Supplementary Table 3).

**Mutational signatures are modulated by risk factors**. In total 10 base-substitution signatures have been identified and referenced so far in liver cancer according to COSMIC nomenclature[3, 11, 16]: signatures 1 and 5 related to aging, 4 to tobacco, 6 to mismatch repair deficiency, 22 to aristolochic acid, 24 to aflatoxin B1 and the remaining signatures 12, 16, 17, and 23 of unknown etiology (Fig. 1a). In our series, mutational analysis using BayesNMF[17] and EMu[18] revealed two new signatures, characterized by T>C mutations in NTT (signature N1) and NTA (signature N2) contexts (Supplementary Fig. 1), but they were each operative in a single patient and require further characterization in additional cases. To correlate signature intensities with clinical and genomic features, we quantified the contribution of the 10 COSMIC signatures in a combined WGS data set comprising our 35 HCC and 264 HCC from the ICGC-Japan series (ICGC-JP), mostly related to HBV (30%) and HCV (56%) infection (Supplementary Table 4)[15]. Signatures 6, 17, 22, 23, and 24 were each operative in ≤5% of cases (Fig. 1b). By contrast, signatures 1, 4, 5, 12, and 16

were operative in most samples and altogether account for 97% of mutations per HCC. These signatures also predominate in HCA and FLC with a stronger contribution of signature 5 (Supplementary Fig. 2). Signatures 1 and 5 are ubiquitous in human cancers and generally correlated with age, although the association is less clear in liver cancers[19] and was not significant in our combined series. By contrast, we found strong correlation with age for signatures 4 ($P = 1.7 \times 10^{-5}$, linear regression model), 12 ($P = 0.019$, linear regression model) and 16 ($P = 4.4 \times 10^{-5}$, linear regression model). Signature 4, highly prevalent in lung cancers from smoker patients, contributed on average 1397 mutations in the liver cancer genomes of smokers but also 1004 mutations in non-smokers ($P = 0.019$, linear regression model, Supplementary Table 5). Thus, tobacco is likely one cause of signature 4 in the liver, but other environmental sources of polycyclic aromatic hydrocarbons (PAH)[20] may contribute to this signature in non-smokers. Consistently, the amount of signature 4 increased with age in both smokers and non-smokers, but contributed only 18 mutations/year in non-smokers vs. 44 mutations/year in smokers (Fig. 1c). Signatures 12 and 16 were so far described exclusively in liver cancers. Signature 12 was strongly enriched in the ICGC-JP cohort, which could be due to geographical, etiological, and/or technical differences. Signature 16 was significantly associated with male gender ($P = 1.5 \times 10^{-6}$, linear regression model), alcohol ($P = 2.0 \times 10^{-6}$, linear regression model), and tobacco consumption ($P = 4.8 \times 10^{-5}$, linear regression model). Similarly to signature 4, the amount of mutations related to signature 16 increased with age but with a dramatically steeper slope in males and alcohol drinkers (Fig. 1d). Thus, the processes generating signatures 4 and 16 are operative in most liver cancers but their activity is modulated by risk factors (age, alcohol, tobacco, and gender). Finally, correlations with clinicopathological feature revealed an increase of signature 5 with higher Edmonson grade (poorly differentiated tumors, $P = 0.0067$, linear regression model) and a positive correlation between signature 16 and tumor size ($P = 0.0066$, linear regression model, Supplementary Fig. 3).

**Mutational signatures and driver genes**. To determine the mutational processes most likely at the origin of driver mutations, we estimated the probability of each individual mutation being due to each process considering the mutation category (substitution type and trinucleotide context) and the proportion of each mutational signature in the tumor genome (see Online Methods). Overall, we observed the same diversity of processes in driver mutations as in other coding mutations, but with differences between genes (Fig. 1e). In particular, *CTNNB1* mutations were significantly more often attributed to mutational signature 16 (median probability = 0.51) as compared with other coding mutations in *CTNNB1*-mutated (0.44, $P = 0.0019$, Wilcoxon rank-sum test) or in *CTNNB1*-wild-type tumors (0.35, $P = 3.3 \times 10^{-10}$, Wilcoxon rank-sum test, Fig. 1f). Signature 16 was dominant across most *CTNNB1* hotspots except on amino acids T41 and S45 where mutations were more often attributed to signature 12 (Fig. 1g). Given that drinkers accumulate twice more signature 16 mutations per year than non-drinkers, we suggest that at least part of the association of *CTNNB1* mutations with alcohol[11] is due to the higher propensity of signature 16 to target *CTNNB1* as compared with other mutational processes operative in liver cancers. In this line, we validated the association of signature 16 with male gender, alcohol and *CTNNB1* mutations in an independent whole exome sequencing (WES) cohort of 573 tumors, comprising TCGA data[21] and our previously published series[11] (Supplementary Fig. 4).

**Impact of DNA replication and transcription on mutational signatures**. We investigated how mutational signatures are

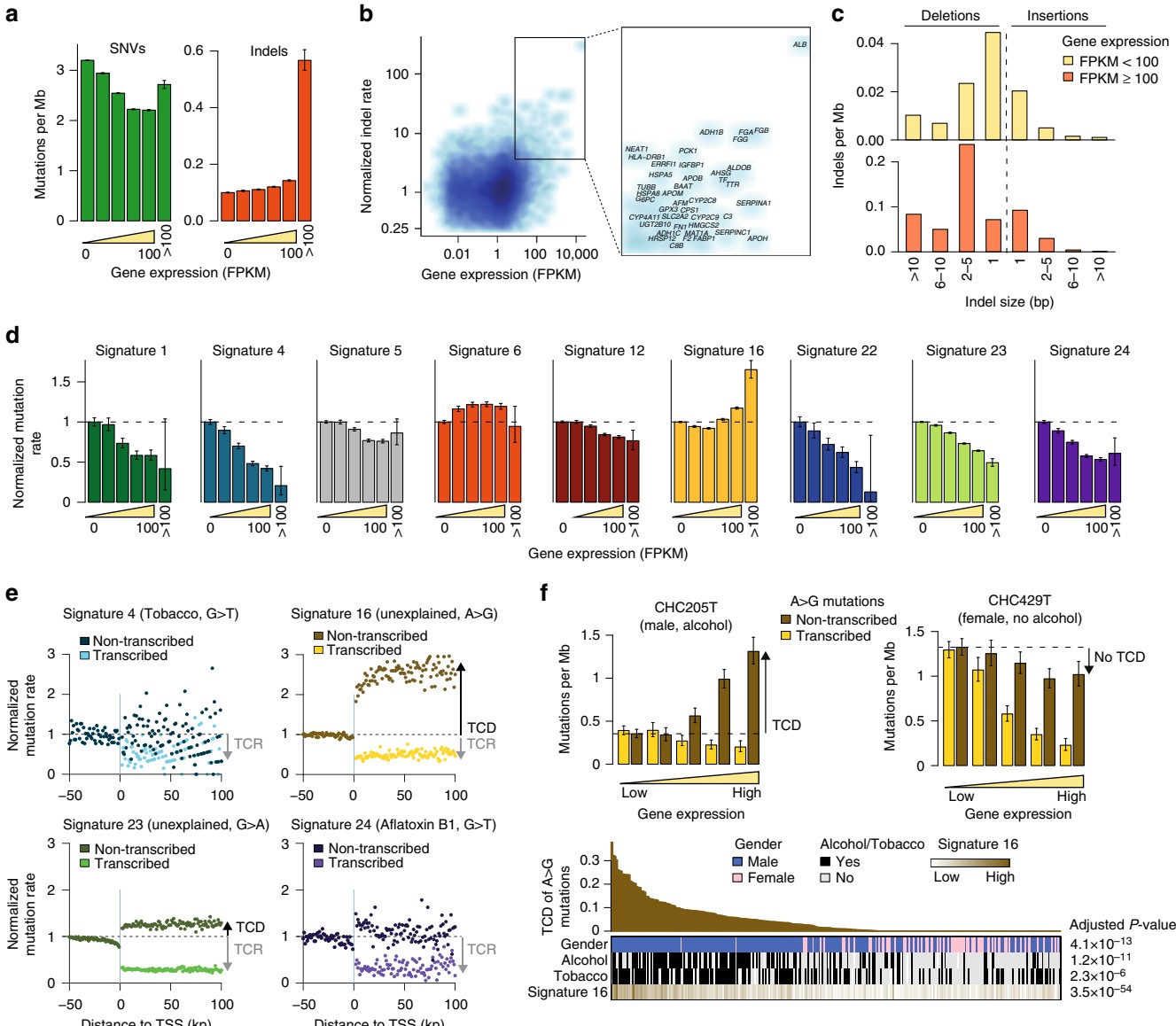

**Fig. 2** Gene expression strongly modulates mutational processes in liver cancer. **a** Number of single nucleotide variants (SNVs, left) and small insertions and deletions (indels, right) per megabase in genes as a function of expression level. Genes with expression between 0 and 100 fragments per kilobase of exons per million reads (FPKM) were divided in 5 gene expression quintiles. A separate group was created for very highly expressed genes (FPKM ≥ 100). Error bars indicate the 95% confidence intervals of the estimated mutation rates. **b** Per-gene correlation of indel rate with gene expression. The normalized indel rate represents the number of indels per megabase in each gene divided by the indel rate in unexpressed genes (FPKM = 0). Very highly expressed genes with a high indel rate are represented in a zoomed box on the right. **c** Distribution of indel types and sizes in very highly expressed (FPKM ≥ 100) vs. all other genes. **d** Correlation between gene expression and SNV rates broken down by mutational signature. Mutation rates were normalized against the mutation rate in unexpressed genes. **e** Evolution of mutation rates when crossing transcription start sites (TSS) for the 4 mutational signatures with the strongest transcriptional strand bias. Each dot represents the average mutation rate in a 1-kb window between 50 kb upstream and 100 kb downstream transcription start sites, normalized against the mutation rate in intergenic regions. In transcribed regions, the mutation rate was estimated separately on the transcibed (light) and non-transcribed (dark) DNA strands. Black and gray arrows represent the shifts in mutation rate attributed to transcription-coupled repair (TCR) and transcription-coupled damage (TCD). **f** Transcription-coupled damage was quantified in each tumor as the increase of A>G mutations between low and high expression genes (top). Bottom: Correlation of TCD with clinical and molecular features

modulated by replication and transcription. As expected, the mutation rate globally increased in late-replicating regions, but with a distinctive gradient for each signature, signatures 4, 6, 12, 16, 23, and 24 displaying subtle but significant asymmetries between the leading and lagging DNA replication strands (Supplementary Fig. 5). Also consistent with previous reports, the mutation rate globally decreased with gene expression. However, we identified an unsuspected rise of indels, and SNVs to a lesser extent, in very highly expressed genes (FPKM>100, Fig. 2a)

including many liver-specific genes like *ALB*, *APOB*, cytochrome P450 enzymes, alcohol dehydrogenase, and fibrinogen subunits (Fig. 2b). Mutations in these genes were strongly enriched in indels (30% in *ALB* and *APOB* vs 3% in other genes, $P = 2.0 \times 10^{-11}$), in particular deletions of 2–5 bases occurring at polynucleotide repeats (Fig. 2c). These indels are characteristic of replication slippage errors and may result from conflicts between the replication and transcription machineries, as demonstrated in vitro[22]. We validated the high amount of indels, in particular

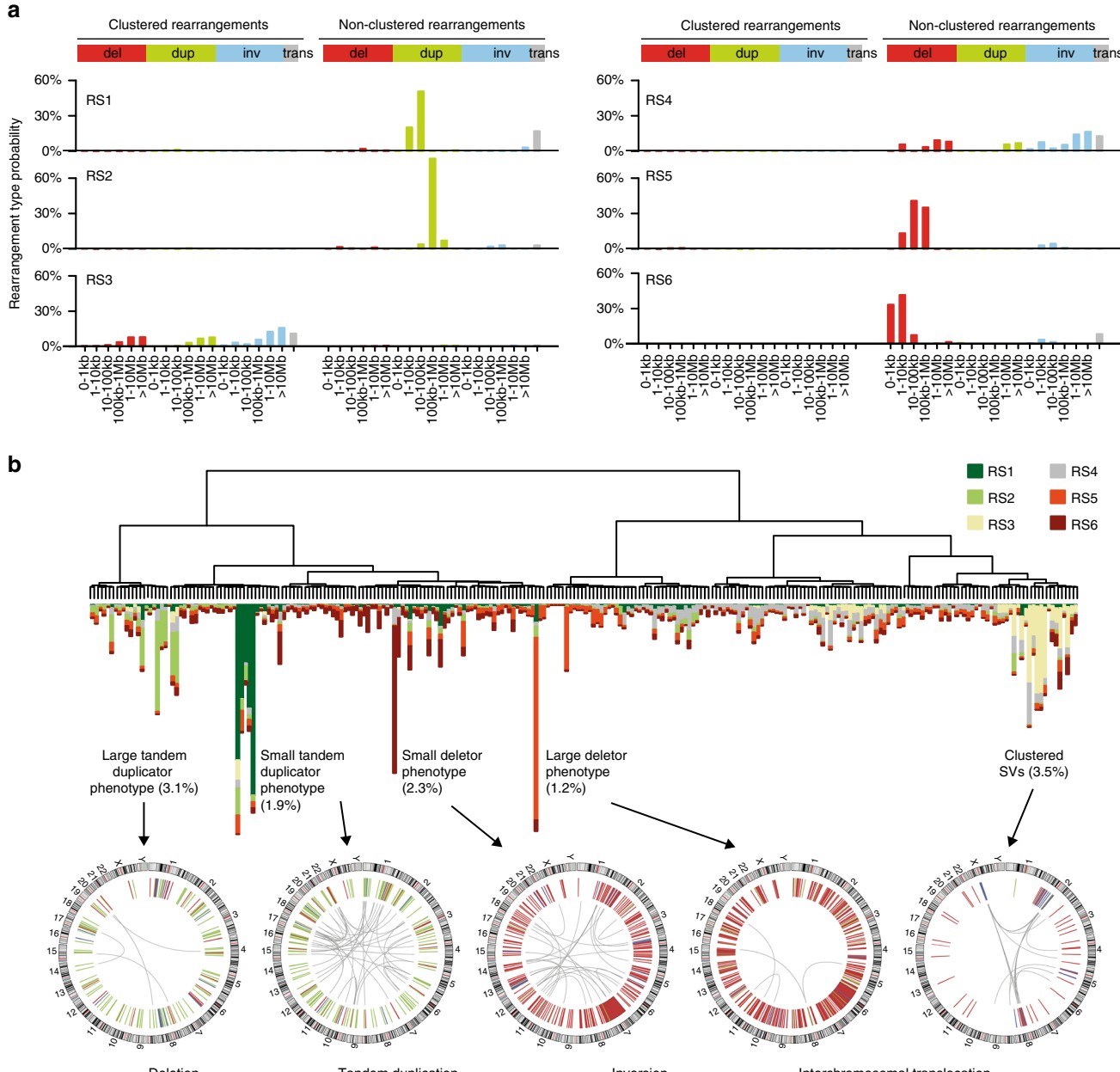

**Fig. 3** Structural rearrangement signatures in liver cancer. **a** Six rearrangement signatures identified by non-negative matrix factorization. Structural rearrangements were classified in 38 categories considering their type (del: deletion, dup: tandem duplication, inv: inversion, trans: interchromosomal translocation) and size, and distinguishing clustered from non-clustered events. The probability of each rearrangement category in each signature is represented, with rearrangement types indicated above and rearrangement sizes below. **b** Unsupervised classification reveals HCC subgroups with similar rearrangement signatures. Particular phenotypes, defined by the presence of ≥50 rearrangements attributed to a same signature, are indicated, with the percentage of tumors displaying the phenotype in parenthesis. Below, CIRCOS plots represent the structural rearrangement profiles of 5 tumors representative of particular structural rearrangement phenotypes

deletions of 2–5 bases, in very highly expressed genes in an independent series of 573 tumors analyzed by WES (Supplementary Fig. 6). Interestingly, the gradient of SNV rate with gene expression differed strongly between mutational signatures (Fig. 2d), signature 16 showing an unsuspected increase in highly expressed genes as well as the strongest transcriptional strand bias (Supplementary Fig. 7). Next, we showed that signature 16, and signature 23 to a lesser extent, were associated with a strong transcription-coupled damage (TCD), characterized by an excess of mutations on the non-transcribed strand as compared to neighboring intergenic regions[6] (Fig. 2e). We quantified TCD in each tumor (Fig. 2f). The magnitude of TCD across patients was higher in males ($P = 4.1 \times 10^{-13}$, Wilcoxon rank-sum test), drinkers ($P = 1.2 \times 10^{-11}$, Wilcoxon rank-sum test) and smokers ($P = 2.3 \times 10^{-6}$, Wilcoxon rank-sum test), and correlated with signature 16 activity ($P = 3.5 \times 10^{-54}$, Pearson correlation test). These findings identify transcription-coupled damage as a hallmark of signature 16, associated with alcohol consumption, leading to an increased mutation rate in highly expressed genes. By contrast, signatures related to carcinogens forming bulky DNA adducts (signatures 4, 22 and 24) displayed a strong depletion of mutations in highly expressed genes and transcriptional strand biases related to transcription-coupled repair (TCR). Altogether, these results highlight the major role of transcription in shaping

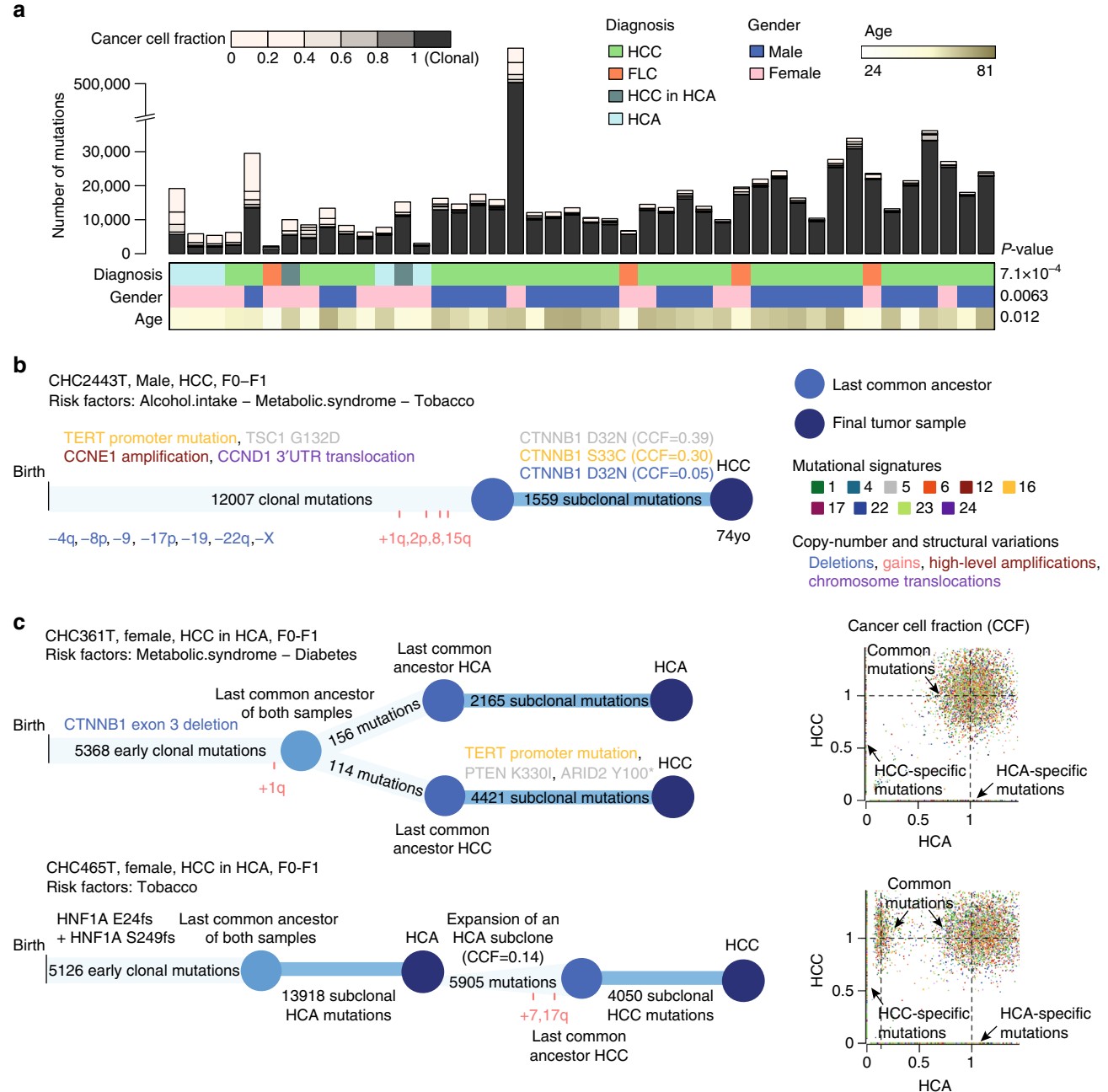

**Fig. 4** Clonal architecture and natural history of liver tumors. **a** Number of clonal and subclonal mutations identified in 44 liver tumors. A color code indicates the number of mutations with a cancer cell fraction (proportion of tumor cells harboring the mutation) between 0 and 1 (clonal mutations). Tumors are ordered according to their proportion of subclonal mutations and significantly associated clinical features are represented below. **b** Clonal history of an hepatocellular carcinoma displaying 3 different subclonal mutations of *CTNNB1*. Driver mutations and copy-number alterations in the clonal and subclonal compartments are indicated with a color code indicating the type of event and the most likely signature of origin, as represented in the legend. Copy-number alterations are indicated and duplications are positioned according to their timing in point mutation time. **c** Clonal histories of two cases of adenoma-to-carcinoma progressions. In CHC361T (*CTNNB1*-related adenoma), few clonal mutations distinguish the adenoma (HCA) and carcinoma (HCC) samples, but 3 subclonal driver mutations were acquired in the carcinoma part. In CHC465T (*HNF1A*-related adenoma), the carcinoma developed from a subclone representing 14% of cells in the adenoma sample that further acquired 5905 mutations. In both cases, chromosome duplications occurred shortly before the carcinoma was operated

liver cancer genomes, in interaction with both endogenous molecular processes and exogenous exposures to genotoxic agents.

**Structural variant signatures define highly rearranged HCC subsets.** The number and type of structural rearrangements varied dramatically between tumors, suggesting the implication of

distinct mutational processes. To identify signatures of these processes, we classified rearrangements in 38 subclasses considering their type and size (see Online Methods), and we applied the same statistical framework used for mutational signatures. In our combined WGS data set, this strategy revealed 6 signatures (RS1 to RS6, Fig. 3a, Supplementary Fig. 8), operative at low levels in most tumors but highly active in rare HCC subsets displaying

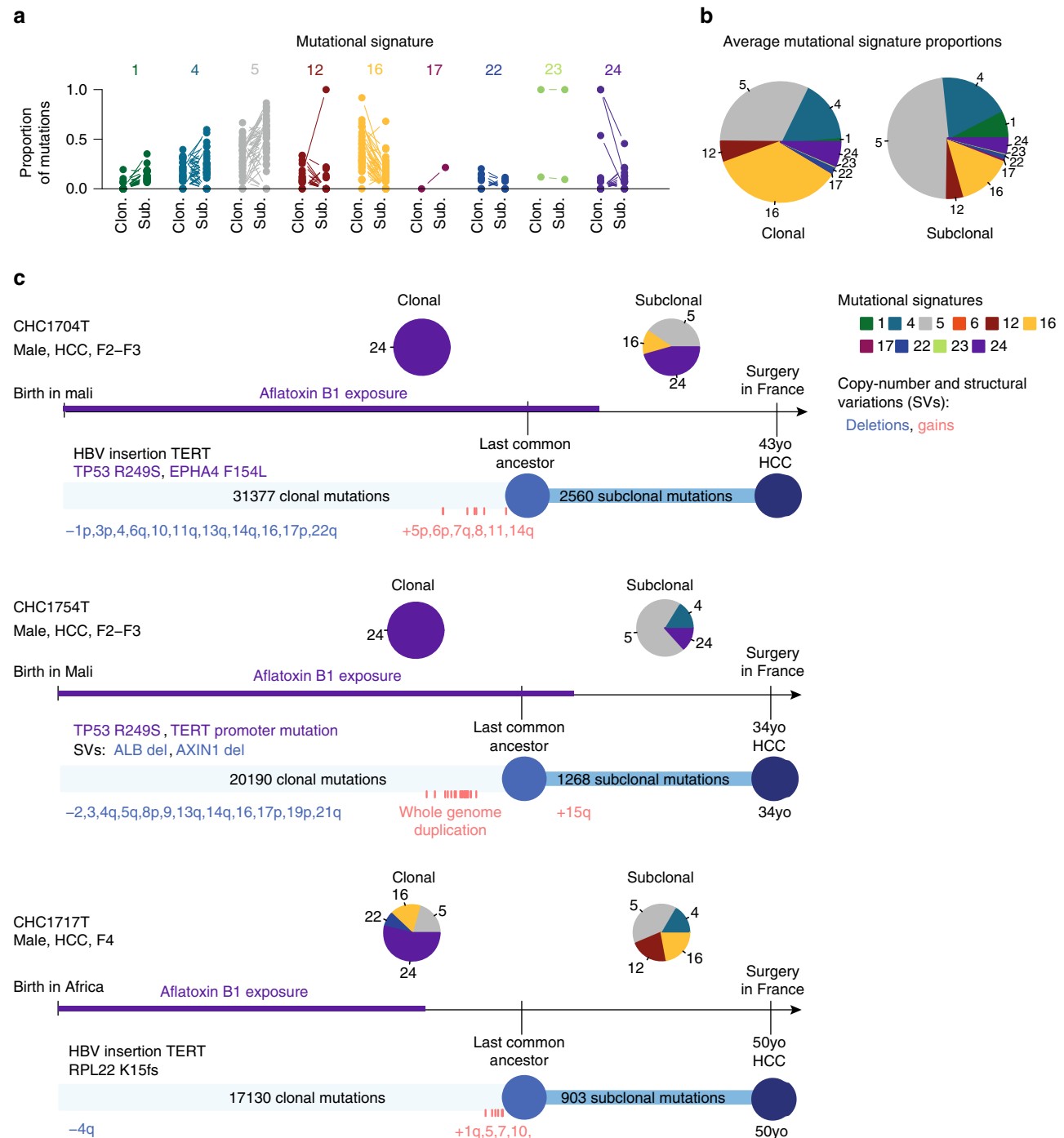

**Fig. 5** Mutational processes evolve along liver tumorigenesis. **a** Proportion of mutations attributed to each mutational signature in the clonal and subclonal mutations of each tumor (connected with a line). Clon. clonal; Sub. subclonal. **b** Average contribution of each signature to clonal and subclonal mutations in our series of 44 liver tumors. **c** Clonal history of 3 HCC developed in African migrants with aflatoxin B1 exposure and HBV infection. Driver alterations in the clonal and subclonal compartments are indicated with a color code describing the type of event and the most likely signature of origin, as represented in the legend. The R249s mutation of *TP53*, hallmark of aflatoxin B1 exposure, is encountered in two patients. *TERT* promoter is altered by mutation in one patient and HBV insertion in two patients. The two *TP53*-mutated tumors display many chromosome deletions, and the 3 cases show an accumulation of chromosome gains shortly before the last selective sweep. Signature 24 (aflatoxin B1) is dominant in clonal mutations but vanishes in subclonal mutations

striking structural rearrangement phenotypes (Fig. 3b). Tandem duplication signatures RS1 and RS2 were characterized by small (<100 kb, RS1) and large (>100 kb, RS2) duplications. Similar signatures were recently described in breast cancers[23, 24] and were associated with *BRCA1* inactivation (small duplications) and *TP53* mutations (large duplications). We found a consistent increase of signature RS2 in *TP53*-mutated tumors ($P = 0.0028$, linear regression model), but signature RS1 was not associated with *BRCA1* inactivation in HCC. Signatures RS5 and RS6 were respectively dominated by large (>10 kb) and small (<10 kb) deletions, signature RS6 showing a modest association with alcohol intake ($P = 0.018$, linear regression model). Signature RS3

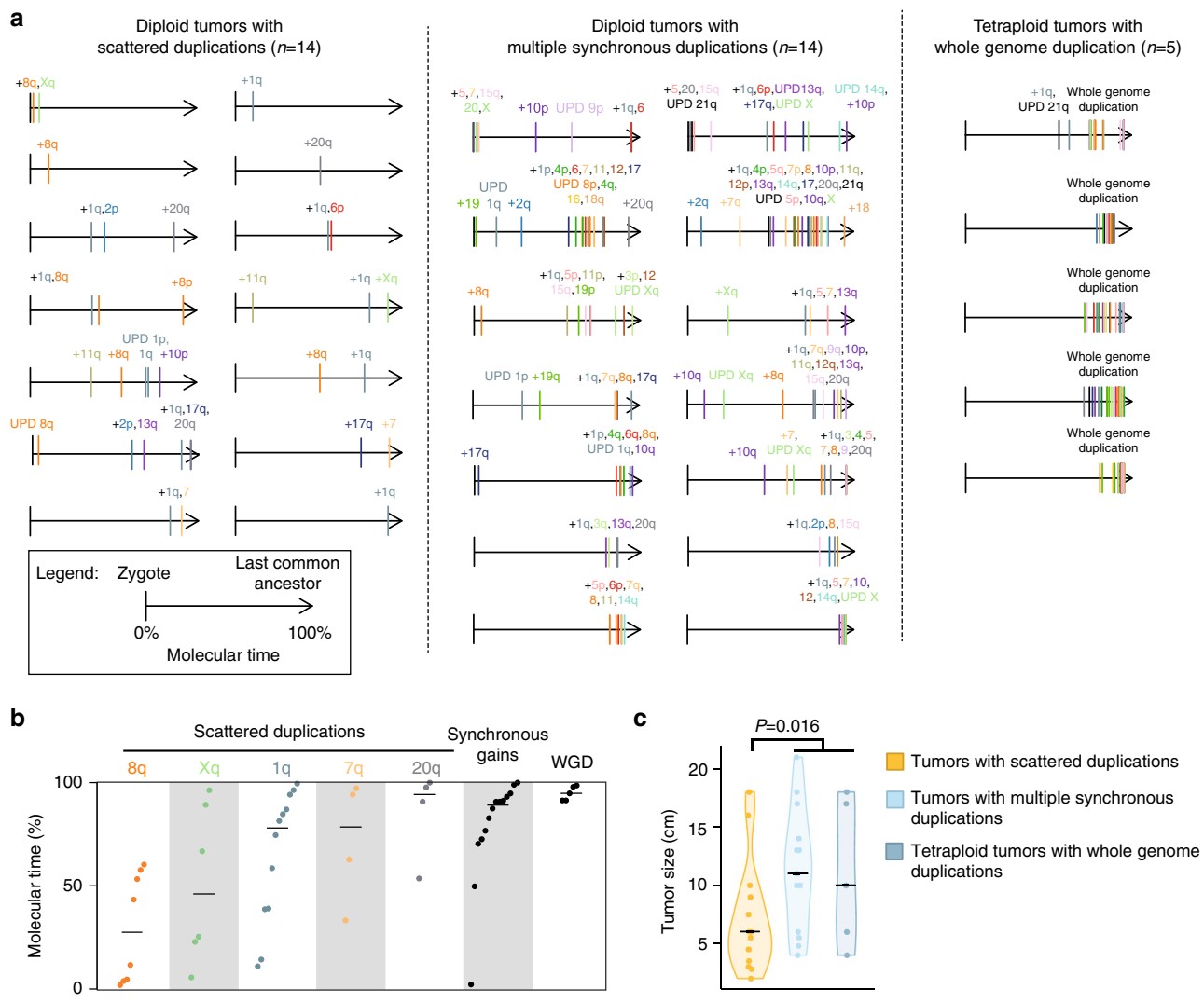

**Fig. 6** Chromosome duplications are late events in hepatocellular carcinomas. **a** Timing of copy-number gains in 33 informative HCC. The timing of chromosome gains was estimated from the ratio of duplicated over non-duplicated mutations. Each arrow represents the point mutation time, 0 being the time when no mutation had been acquired yet and 100 the time when all clonal mutations were present. Chromosome gains are placed on this scale with a color code for each chromosome. Tumors are grouped based on the presence in their history of whole genome duplications or multiple synchronous gains defined as the co-occurrence of ≥4 duplications in a window shorter than 30% mutation time. When several copies of a same chromosome were gained, only the first event is represented for clarity. **b** Distribution of duplication timings in point mutation time for the most recurrent scattered duplications, synchronous gains of ≥4 chromosome regions and whole genome duplications. **c** Tumor size distribution in tumor groups showing different patterns of chromosome duplication acquisition, as defined in **a**

was characterized by clustered rearrangements, often associated with complex amplification events, whereas signature RS4 was dominated by non-clustered inversions and translocations and associated with higher Edmonson grade ($P = 0.0017$, linear regression model). Structural variants were globally enriched in early-replicating and highly transcribed regions, duplication signatures RS1 and RS2 showing the strongest association with replicative and transcriptional contexts (Supplementary Fig. 9).

**Clonal architecture and timing of driver mutations**. The sequencing coverage in our series (92-fold) allowed us to identify subclonal mutations and copy-number alterations (CNA) and to reconstruct the natural history of each tumor (Supplementary Data 2). We identified a median of 1953 subclonal mutations in each sample (range 780–92,279), most of which were present in <20% of cancer cells (Fig. 4a). The proportion of subclonal mutations was significantly higher in adenomas ($P = 0.012$,

Wilcoxon rank-sum test), females ($P = 0.0063$, Wilcoxon rank-sum test) and younger patients ($P = 0.012$, Pearson correlation test), suggesting that these tumors are more heterogeneous than more classical HCC. Most driver mutations were clonal but subclonal mutations in CTNNB1, AXIN1, PTEN (×2), RB1 and ARID2 were identified in 5 tumors (Supplementary Table 6). In one of these HCC, 3 different subclonal mutations of CTNNB1 (cancer cell fraction = 5, 30, and 39%) affected the D32 and S33 hotspots (Fig. 4b) suggesting that the WNT/ß-catenin pathway activation was selected independently in several clones late during tumor progression in this patient. We previously showed that TERT promoter mutations are early events occurring in pre-malignant HCC lesions whereas other driver mutations, including CTNNB1, occur in later stages[25, 26]. By contrast, CTNNB1 is frequently mutated in benign adenomas at risk of malignant transformation requiring in most of cases TERT promoter mutations for carcinoma development[27, 28]. Here, we analyzed two cases of adenoma to carcinoma progression by WGS

(Fig. 4c). One carcinoma displayed subclonal acquisition of *TERT* promoter, *ARID2*, and *PTEN* mutations in 13% of tumor cells. The other carcinoma progressed from a subclone representing 14% of the adenoma sample, without new driver gene mutation but with acquired duplications of chromosome 7 and 17q. These results show that tumor heterogeneity is more important in benign liver tumors, with carcinoma progression originating from one subclone, and, in line with our previous findings, suggest that functional selection of β-catenin activation may occur at different steps of hepatocarcinogenesis.

**Mutational signature evolution along tumorigenesis.** Mutational signatures evolved between clonal and subclonal mutations, with a general trend towards a decrease of signature 16 and an increase of signatures 1 and 5 (Fig. 5a, b). The most striking evolution was observed in 3 African migrants who displayed high contribution of the aflatoxin B1-related signature 24 in clonal mutations that strongly decreased or disappeared in subclonal mutations (Fig. 5c). These patients arrived in France several years before surgery. Thus, most clonal mutations were acquired when the patients were in Africa. In contrast, most of the subclonal diversification occurred when the patients were in France, not exposed to aflatoxin B1 anymore. The natural histories reconstructed for the 44 tumors in our series (Supplementary Data 2) reflect the high molecular diversity of liver tumors.

**Chromosome duplication timing.** Chromosome or whole genome duplications are frequent in liver cancers (Supplementary Fig. 10), but whether these events occur early or late during tumor progression is unknown. We used the proportion of duplicated mutations to estimate the molecular timing of each duplication in our series or 35 HCC, as previously described[29]. Fourteen tumors displayed scattered duplications (Fig. 6a). These duplications occurred earlier during tumor development (median timing = 60% point mutation time, pmt), in particular 8q and Xq gains (Fig. 6b), 8q gains being often acquired and selected repeatedly along tumorigenesis (Supplementary Fig. 11). By contrast, synchronous gains observed in 14 HCC (89% pmt), and whole genome duplications found in 5 tetraploid HCC (95% pmt) were very late events associated with bigger tumors (median size 11 cm vs 6 cm in other tumors, $P = 0.016$, Wilcoxon rank-sum test, Fig. 6c) of higher Edmonson grade ($P = 0.017$, Chi-square test for trend), and may represent rate-limiting steps allowing the expansion of the tumor to a detectable mass.

## Discussion

This comprehensive study of >300 tumor genomes revealed that 10 mutational signatures account for almost all mutations in liver cancers. Other signatures probably remain to be identified, but they will represent rare processes operative in a very low proportion of patients. In our series of 44 liver cancers enriched in cases developed in absence of cirrhosis and known etiology, we identified 2 new signatures operative in a single patient. Signature N1, characterized by T > C at NTT site, was encountered in a 67 year-old woman who developed an adenoma with carcinoma transformation on a non-cirrhotic liver with metabolic syndrome. Both the adenoma and carcinoma counterparts displayed this signature. Signature N2, characterized by T > C mutations at NTA sites, was encountered in a 65 year-old woman with non-cirrhotic liver and unknown etiology. These signatures may represent rare mutational processes that remain to be explained.

A major finding of this study is that a combination of 5 ubiquitous signatures (COSMIC signatures 1, 4, 5, 12, 16) accounts on average for 97% of mutations in an HCC genome. Signatures 1 and 5 are known clock-like signatures of mutational processes

operative in all tissues[19]. Signature 1 accounts for the spontaneous deamination of methylated cytidines at CpG sites and is predominant in tissues with a high rate of cell divisions, for example stomach and colorectum. The process underlying signature 5 is still unclear but the associated transcriptional strand bias suggests that it may be due to a ubiquitous metabolic mutagen. A recent analysis of adult stem cells showed that signature 5 was predominant over signature 1 in the liver, accounting for around 1000 mutations per cell in a 50 year-old individual[30], consistent with the present report. Signature 4 has been clearly associated with direct tobacco smoke exposure in lung and larynx cancers[31], and is very similllar to the mutational signature induced in vitro by exposing cells to benzo[a]pyrene[32]. Consistently, we found a more than two-fold increase in the number of mutations due to signature 4 accumulated per year in liver cancers from smokers as compared with non-smokers. However, a substantial amount of mutations due to signature 4 was detected in non-smokers, contrary to lung cancers where this signature is virtually absent in non-smokers. Thus, other sources of PAH may generate signature 4 mutations in the liver[20, 33], a process that would be accelerated by smoking. Finally, signature 16 is the real hallmark of liver cancers, operative in every tumor and accounting on average for 40% of all somatic mutations. We found a strong increase of this signature independently associated with male gender, alcohol and tobacco consumption. Interestingly, a recent analysis of 94 esophageal squamous-cell carcinoma revealed a similar signature associated with alcohol consumption[34]. Although the molecular mechanism generating these mutations remains to be elucidated, these data and the strong transcriptional strand bias associated with signature 16 suggest that it may be due to a product of liver metabolism forming bulky DNA adducts on adenine residues, which production would be increased in males, smokers and drinkers.

In addition to the 5 ubiquitous signatures, we describe 5 sporadic signatures, some of which are operative in a very low number of patients but may account for an extremely high number of mutations: >600,000 mutations for signature 23 in tumor CHC892T. This signature, together with the mismatch repair deficiency signature 6, highlight rare etiologies of HCC leading to hypermutated tumors that may benefit from immunotherapy[35]. Signatures 22 (aristolochic acid) and 24 (aflatoxin B1) were relatively rare in our series from patients treated in France or Japan, but may be extremely frequent in other geographical areas, examplified by a recent report on aflatoxin B1-related HCC in China[36].

Our signature-based analysis revealed very diverse effects of replication and transcription on mutation rates and strand asymmetries. The most striking finding was the strong transcription-coupled damage associated with male gender, alcohol and tobacco consumption. Haradhvala et al.[6] reported for the first time this process associated with T > C mutations in liver cancers. Here we show that transcription-coupled damage is a specific feature of mutational signature 16. We also identified a new mechanism generating numerous indels in very highly expressed genes. Imielinski et al.[37] recently described indel hotspots in lineage-defining genes enriched at specific sequence motifs and chromatin contexts. Our data expand these findings in the liver and identify transcription as the main determinant of this indel-generating mechanism, operative in genes expressed with an FPKM > 100 and remarkably correlated with expression level. We[11] and others[21] previously underlined the high number of mutations in hepatic differentiation genes. These new findings suggest that hepatic differentiation genes may be particularly prone to indel accumulation due to their high expression levels rather than selected because they provide a proliferative advantage to tumor cells.

We extended our signature analysis beyond point mutations to explore structural rearrangement signatures. Strikingly, the 6 rearrangement signatures that we identified in liver cancers are very similar to the 6 signatures identified in breast cancers[23]. In particular, we identified two signatures characterized by small (<100 kb) and large (between 100 kb and 1 Mb) tandem duplications, reminiscent of the tandem duplication signatures associated with homologous recombination deficiency in breast cancers[24]. Although the molecular causes of these phenotypes in HCC remain to be identified, these findings suggest that the molecular mechanisms underlying structural rearrangement may be more general than mutational processes.

Finally, we explored the clonal architecture of liver tumors. We identified subclones in every sample that generally represented <20% of the sample's tumor cells, benign tumors being more heterogeneous than classical HCC. We reconstructed the natural history of each tumor, showing that the order in which driver events occur is not very constrained: a same driver mutation or copy-number alteration can be early in one tumor and late in another. However, general rules emerge, like the diminution of signature 16 in late subclonal mutations or the late acquisition of chromosome gains. The latter is in agreement with a recent pan-cancer analysis showing that chromosomal gains are typically acquired during the second half of clonal evolution[38]. In particular, we identified several modes of duplication acquisition, multiple synchronous gains and whole genome duplications being very late events associated with larger tumors.

A limitation of this study is the small sample size for some etiologies. By grouping our new WGS data set with the ICGC-Japan series, we obtained substantial sample sizes for the main etiologies in France and Japan: alcohol, HBV and HCV infection. However, other etiologies like metabolic syndrome and hemochromatosis were poorly represented. Future whole genome studies will need to include more samples from other geographic areas associated with different risk factors, including nonalcoholic steatohepatitis that accounts for the rise of HCC cases in the US[39].

Altogether, our study sheds new light on the diversity of processes generating somatic mutations, their interactions with risk factors, cellular processes and driver genes, and their evolution along hepatocarcinogenesis.

## Methods

**Clinical samples.** A series of 44 liver tumor samples and their non-tumor counterparts were collected from patients surgically treated in four French hospitals located in Bordeaux and Paris region. The study was approved by institutional review board committees (CCPRB Paris Saint-Louis, 1997, 2004 and 2010, approval number 01–037; Bordeaux, 2010-A00498-31). Written informed consent was obtained in accordance with French legislation. All samples were immediately frozen in liquid nitrogen and stored at −80 °C. Tumors included in this study comprised 35 hepatocellular carcinomas (HCC), 3 hepatocellular adenomas (HCA), 4 fibrolamellar carcinomas (FLC) and 2 cases of HCA to HCC progressions with both HCA and HCC samples. HCC were enriched in cases developed on a non-cirrhotic liver (32/35, 91%) to unmask traces left by exogenous toxic exposures: 25 tumors developed in non-fibrotic (METAVIR F0-F1), 7 in chronic hepatitis (F2-F3) and 3 in cirrhotic liver (F4). Besides, 15 patients belonged to the national NoFlic cohort dedicated to the identification of new etiologies in non-fibrotic HCC cases. Clinicopathological data were available for all cases. Risk factors were defined by substantial alcohol intake, HCV, HBV, hemochromatosis and metabolic syndrome. Metabolic syndrome consists of a combination of disorders, including central obesity (waist circumference >102 cm (males), >88 cm (females)), hypertriglyceridemia (triglycerides >150 mg/dl), low high-density lipoprotein serum levels (<40 mg/dl), arterial hypertension (>130 mm Hg systolic or >85 mm Hg diastolic) and raised fasting plasma glucose (FPG) levels 1.1 mg/dl or previously diagnosed type 2 diabetes. At least three of the latter criteria had to be fulfilled for diagnosis. Patients without known etiology were those that did not display any of the above frequent etiologies or rare etiologies (such as primary biliary cirrhosis, autoimmune hepatitis and primary sclerosing cholangitis). A diversity of risk factors were represented in our series, including HBV ($n = 5$), HCV ($n = 4$), alcohol ($n = 12$), metabolic syndrome ($n = 7$), hemochromatosis ($n = 1$) and without etiology ($n = 6$). Detailed clinical characteristics of each sample are provided in Supplementary Table 1.

**Whole genome sequencing.** We extracted DNA using a salting-out procedure[40]. Genomic DNA was loaded on a 0.8% agarose gel for quality control; only DNA > 10 kb in size was selected. DNA quantification was performed using Hoechst 33258 from Sigma Chemical. Fourteen pairs (CNG series) were sequenced at the Centre National de Génotypage (Evry, France) and 30 pairs (Integragen series) were sequenced at Integragen (Evry, France). Samples of the CNG series were sequenced on an Illumina Genome Analyzer as paired-end 75, 95, or 100 base pair (bp) reads and on an Illumina HiSeq as paired-end 100 bp reads. Samples sequenced at Integragen were sequenced on an Illumina Hiseq 2500 as paired-end 125 bp reads. Sequences were aligned to the hg19 version of the human genome using BWA[41] version 0.7.15. We used Picard tools version 1.108 (http://broadinstitute.github.io/picard/) to remove PCR duplicates and GATK[42] version v3.5 for local indel realignment and base quality recalibration, as recommended in GATK best practices[43]. We obtained an average depth of 92-fold for tumors (range 63–109) and 67-fold for matched non-tumor liver samples (range 35–116). Coverage and variant calling statistics for each sample are provided in Supplementary Table 2.

**Somatic mutation calling.** We created a panel of normals (PON file) using GATK and we used MuTect2 to call somatic mutations (single nucleotide variants and small insertions and deletions) by comparing each tumor sample with its matched non-tumor counterpart and the PON file. Based on visual inspection of 188 SNVs and 76 indels using the Integrative Genomics Viewer (IGV)[44], we established the following post-filtering criteria. First we selected mutations covered by ≥6 reads in both the tumor and non-tumor samples, with <5% of variant reads in the non-tumor sample and we excluded mutations belonging to the ENCODE Data Analysis Consortium blacklisted regions (http://hgdownload.cse.ucsc.edu/goldenPath/hg19/encodeDCC/wgEncodeMapability/wgEncodeDacMapabilityConsensusExcludable.bed.gz). We then processed differently SNVs and indels. For SNVs, we selected mutations with a MuTect2 filter flag among "PASS", "clustered_events" or "t_lod_fstar". To improve specificity in the calling of mutations with low variant allele frequencies (VAF < 0.2), we quantified the number of high quality variant reads in the tumor (mapping quality ≥50, base quality ≥30) and the number of variant reads in the non-tumor sample with no quality threshold using bamreadcount (https://github.com/genome/bam-readcount). Only variants supported by ≥2 high quality reads in the tumor (≥3 for samples of the CNG series with higher coverage) and no variant read in the matched non-tumor sample were selected. For indels, we selected mutations with a MuTect2 filter flag among "PASS", "clustered_events" or "str_contraction" supported by ≥ 20% reads in the tumor sample.

**Copy-number and structural variant analysis.** We used MANTA[45] software to identify somatic structural variations (Supplementary Data 1) and viral insertions (Supplementary Table 3) from the tumor and non-tumor bam files. To keep only the most reliable events, we selected only structural variants (SVs) supported by ≥ 15 reads (for all SV types), representing ≥ 10% of reads (for inversions and interchromosomal translocations). We used cgpBattenberg[29] algorithm to reconstruct copy-number profiles from whole genome sequences of our 44 liver tumors. This method uses both coverage data and allele frequencies of germline SNPs to estimate the tumor cell content, ploidy and absolute copy-numbers of each allele across the genome. Besides, germline SNPs are phased and haplotype-based allelic frequencies are used to identify subclonal copy-number alterations.

**Mutational signature analysis.** Ten mutational signatures were described so far in liver cancers[3, 11, 16] and are referenced in COSMIC database: signatures 1, 4, 5, 6, 12, 16, 17, 22, 23, and 24. To identify new mutational signatures in our series of 44 liver tumors, we first performed a de novo extraction of signatures using BayesNMF[46, 47] and EMu[18] methods. BayesNMF extracted 10 signatures and Emu 9 signatures, which we then compared to the pan-cancer catalog of 30 signatures referenced in COSMIC database. The comparison was performed using the cosine similarity score, as previously described[3]. Most signatures corresponded to one or a mixture of two already known signatures and displayed a cosine similarity >0.75 with at least one of the ten COSMIC signatures known to be operative in liver cancers. However, two signatures were identified consistently by BayesNMF and EMu that did not correspond to any previously described signature (cosine similarity <0.75). These new signatures were named N1 and N2 (Supplementary Fig. 1). Signature N1, characterized by T > C at NTT site, was encountered in a 67 year-old woman who developed an adenoma with carcinoma transformation on a non-cirrhotic liver with metabolic syndrome. Both the adenoma and carcinoma counterparts displayed this signature. Signature N2, characterized by T>C mutations at NTA sites, was encountered in a 65 year-old woman with non-cirrhotic liver and unknown etiology. These signatures require confirmation in additional cases.

To correlate mutational signatures with clinical annotations and genomic features, we put together a series of 308 liver cancer genomes comprising our series of 44 tumors (35 HCC) and the ICJC Japan series of 264 HCC[15]. We then used the patterns of the 10 known liver cancer signatures (COSMIC signatures 1, 4, 5, 6, 12,

16, 17, 22, 23, and 24) and we estimated the exposure of each sample to each of these mutational processes. The pattern of each signature consists of mutational probabilities for 96 mutation categories, defined by the 6 substitution types multiplied by 16 possible trinucleotide contexts[3]. We downloaded these patterns from COSMIC website to generate a mutational signature matrix $P =$

$$\begin{pmatrix} p_1^1 & \cdots & p_{10}^1 \\ \vdots & \ddots & \vdots \\ p_1^{96} & \cdots & p_{10}^{96} \end{pmatrix}$$ where $p_i^j$ is the probability of the process $i$ to cause a mutation

of category $j$. We represented the mutation catalogs of the 308 tumors as a

mutation matrix $M = \begin{pmatrix} m_1^1 & \cdots & m_{308}^1 \\ \vdots & \ddots & \vdots \\ m_1^{96} & \cdots & m_{308}^{96} \end{pmatrix}$ where $m_i^j$ is the number of mutations

of category $j$ in tumor $i$. We then used non-negative matrix factorization, as implemented in the NMF package[48], to estimate the exposure matrix $E =$

$$\begin{pmatrix} e_1^1 & \cdots & e_{308}^1 \\ \vdots & \ddots & \vdots \\ e_1^{10} & \cdots & e_{308}^{10} \end{pmatrix}$$ where $e_i^j$ is the number of mutations attributed to process $i$ in

tumor $j$. NMF identifies the matrix $E$ that verifies $M \approx P \times E$ and minimizes a Frobenius norm while maintaining non-negativity[2]. Signatures accounting for <6% of a tumor genome were discarded to avoid overfitting[3, 49]. To compare the activity of mutational processes in clonal and subclonal mutations, we computed the exposure matrices $E$ independantly for clonal and subclonal mutations.

**Association of mutational signatures with driver genes**. We estimated the probability of each somatic mutation being due to each mutational process considering the mutation category and the number of mutations attributed to each process in the corresponding tumor. Let us consider a mutation category $c$ out of the 96 mutation categories defined above. The number of mutations of category $c$ in a tumor $t$ can be expressed as:

$$m_t^c = \sum_{s=1}^{10} p_s^c \times e_t^s$$

where the product $p_s^c \times e_t^s$ represents the number of mutations of category $c$ attributed to signature $s$ in tumor $t$. The probability $P(m,s)$ of a mutation $m$ of category $c$ in tumor $t$ being due to signature $s$ can then be estimated as:

$$P(m,s) = \frac{p_s^c \times e_t^s}{\sum_{s=1}^{10} p_s^c \times e_t^s}$$

Let us now consider all the mutations $m$ affecting a gene (or set of genes) $G$, the contribution of a mutational signature $s$ to mutations in $G$ can be estimated as the cumulative probabilities of mutations $m$ affecting gene $G$:

$$P(G,s) = \sum_{m \in G} P(m,s)$$

We used the above formulas to estimate the probability of each mutation being due to each mutational process, and the contribution of each signature to driver genes and *CTNNB1* hotspot mutations. To identify genes preferentially altered by specific mutational processes, we then compared, for each driver gene $G$ and mutational signature $s$, the distribution of probabilities $P(m,s)$ in mutations affecting gene $G$ compared to all other mutations in the series or all other mutations in samples harboring a mutation in gene $G$ using Wilcoxon rank sum tests. We tested all driver genes identified in a previous study[11] with a q-value<0.05 (*CTNNB1, TP53, ARID2, NFE2L2, ARID1A, ACVR2A, AXIN1, KEAP1, RB1, RPS6KA3, ALB, CDKN2A, RPL22,* and *CDKN1A*). Only associations between signature 16 and *CTNNB1* were significant after Benjamini Hochberg correction for multiple testing.

**Rearrangement signature analysis**. We applied the same statistical framework used for mutational signature analysis to structural variants, as previously described[23]. In short, we defined 38 categories of structural variants considering the type (deletion, tandem duplication, inversion, interchromosomal translocation) and size (<1 kb, 1–10 kb, 10–100 kb, 100kb–1 Mb, 1–10 Mb, >10 Mb) of rearrangements. We also considered differently clustered events that may be related to a same mechanism, from non-clustered events. We used bedtools[50] cluster function to identify clustered events, defined by the presence of ≥10 breakpoints within a 1 Mb window. We then used non-negative matrix factorization, as implemented in the NMF package[48], to extract rearrangement signatures and their exposure in each tumor.

**Association of signatures with clinical features**. To identify clinical and molecular features associated with a mutational signature, we first tested the association of each feature through linear regression models including tumor series as second parameter to account for technical and analytical differences between our cohort

and the ICGC-Japan series. All significant features were then included in a multivariant logistic regression analysis to identify independantly significant features.

**Replication timing and replication strand bias**. We used replication sequencing (Repli-seq) data generated by the ENCODE consortium for the liver cancer cell line HepG2 to define early and late-replicating regions as well as leading and lagging DNA strands as previously described[7]. In short, replication timing deciles were defined using wavelet-smoothed Repli-seq signals downloaded from the ENCODE website. We used Repli-Seq signal peaks (replication initiation) and valleys (replication termination) defined by ENCODE to define leading and lagging strands based on the direction of the replication fork. To avoid ambiguous status, only regions with a peak-to-valley distance ≥500 kb were assigned strand information. This allowed to classify unambiguously 1.5 Gb of the human genome.

To estimate mutation rates within each replication timing decile, we divided the number of mutations by the informative genomic size of the decile, excluding 'N' bases. Signature-wise analyses were performed similarly using only mutations attributed to a single signature with a probability ≥0.7. Replicative strand biases were analyzed by comparing the number of mutations occurring on the leading and lagging strands, over the genome or within each replication timing decile.

**Transcription levels and transcriptional strand bias**. To correlate mutational processes with gene expression, we first estimated the median expression level of GENCODE genes (version 24) in an in-house RNA-seq data set of 39 non-tumor liver samples. We used these expression levels to define 5 gene expression categories from no to high transcriptional activity. We created a separate class for very highly expressed genes, defined by an FPKM (fragments per kilobase of exons per million reads) value ≥ 100. To estimate the mutation rate within each gene expression category, we divided the number of mutations in genes belonging to this category (considering exons and introns) by the cumulated size of these genes. Signature-wise analyses were performed similarly using only mutations attributed to a single signature with a probability ≥ 0.7.

Transcriptional strand biases were analyzed by comparing the number of mutations occurring on the transcribed and non-transcribed strands, over the genome or within each gene expression category. To evaluate how transcription modulates mutation rates on each strand, we grouped mutations in 1 kb bins occurring between 50 kb upstream and 100 kb downstream of transcription start sites. We then estimated the mutation rate on the transcribed and non-transcribed strands within each bin for each mutational signature. We also quantified transcription-coupled repair (TCR) and transcriptional-coupled damage (TCD) for A > G mutations in each sample as the slopes of mutation rates between the lowest and highest gene expression quintiles on the transcribed and non-transcribed strand, respectively.

**Clonality and natural history of each tumor**. For each mutation, we calculated the proportion of mutated reads (variant allele fraction, VAF) and we estimated the proportion of tumor cells harboring the mutation (cancer cell fraction, CCF) as previously described[29, 51], using:

$$CCF = VAF \times \frac{\rho N_t + (1-\rho) N_n}{\rho n_{chr}}$$

where $\rho$ is the tumor cell content, $N_t$ and $N_n$ the copy-number at the locus in tumor and normal cells, and $n_{chr}$ the number of chromosomal copies harboring the mutation in tumor cells (also called multiplicity of the mutation). $\rho$ and $N_t$ were estimated using cgpBattenberg algorithm, and $n_{chr}$ was set to the integer value closest to:

$$\max\left(1, VAF \times \frac{\rho N_t + (1-\rho) N_n}{\rho}\right)$$

We determined the 95% confidence of VAF using a binomial test and we converted this interval to the 95% confidence interval of CCF using the above formula. A mutation was considered subclonal if the upper boundary of the 95% confidence interval was <0.95, and clonal otherwise.

We represented the natural history of each tumor as a diagram indicating the number of clonal and subclonal mutations and the driver events on each branch with a color code indicating the type of event and the most likely causal mutational signature for SNVs. Clonal and subclonal copy-number alterations were also indicated, and duplications were timed as described below.

**Timing chromosome duplications**. When a chromosome duplication occurs, e.g., from 2 to 3 copies, the mutations acquired on the duplicated chromosome copy prior to the duplication event are also duplicated. These mutations will have an increased variant allele fraction (VAF = 2/3 in absence of contamination by normal cells) whereas mutations that were present on the non-duplicated copy or that are acquired after the duplication event will have a lower VAF (1/3). Besides, a chromosome duplicated early in tumor history will have few duplicated mutations compared to a chromosome duplicated late in tumor history. We used the number of duplicated and non-duplicated mutations to estimate the timing of each

chromosome duplication in our series of 44 liver tumors, as previously described[29, 52] with some modifications. Let us consider the simple case of a chromosome with absolute copy-number $N_t = 3$. The molecular time at which the extra copy of the chromosome was gained can be estimated as:

$$T = \frac{N_{\text{dup}}}{N_{\text{dup}} + \frac{N_{\text{ndup}} - N_{\text{dup}}}{3}} \times 100$$

where $N_{\text{dup}}$ and $N_{\text{ndup}}$ are the number of duplicated and non-duplicated mutations, respectively. We extrapolated this formula to chromosomes with $N_t \geq 4$. In this case, we timed the first duplication event using:

$$T = \frac{N_{\text{dup}}}{N_{\text{dup}} + \frac{N_{\text{ndup}} - N_{\text{dup}}}{(3 + N_t)/2}} \times 100$$

where $N_{\text{dup}}$ is the number of mutations at the maximal level of multiplicity and $N_{\text{ndup}}$ the number of mutations at intermediate levels of multiplicity or non-duplicated.

For cases where the two parental chromosome copies were duplicated, e.g., $N_t = 4$ with 2 copies of each chromosome copy, we adapted the formula as follows:

$$T = \frac{N_{\text{dup}}/2}{N_{\text{dup}}/2 + \frac{N_{\text{ndup}}}{(3 + N_t)/2}} \times 100$$

We applied these formulas to time every clonal duplication in our series with at least 30 somatic mutations located within the duplicated chromosome region.

**Computing codes**. The functions created to perform this work and generate figures are available as an open-source R package, Palimpsest, available on Github: https://github.com/FunGeST/Palimpsest.

**Data availability**. The sequencing data reported in this paper has been deposited in the EGA (European Genome-phenome Archive) database (accessions EGAS00001002091, EGAS00001002408 and EGAS00001000706) and the International Cancer Genome Consortium (ICGC) data portal (http://dcc.icgc.org/; release 24, April 2017).

**URLs**. R software, v3.2.3, http://www.R-project.org/;
GATK4, http://www.broadinstitute.org/gatk/;
Oncotator, http://www.broadinstitute.org/cancer/cga/Oncotator/;
cgpBattenberg, https://github.com/cancerit/cgpBattenberg;
ICGC data portal, https://dcc.icgc.org/;
COSMIC mutation signatures database, http://cancer.sanger.ac.uk/cosmic/signatures;
COSMIC cancer census genes, http://cancer.sanger.ac.uk/census;
GENCODE v19, http://www.gencodegenes.org/releases/19.html;
ENCODE project, https://www.encodeproject.org/.

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

## Acknowledgements

We thank J.C. Nault for critical review of the manuscript and T. Shibata for helpful discussions on mutational signatures. We also thank the Réseau National Centre de Ressources Biologiques (CRB) Foie, the French Liver Biobanks network (INCa, BB-0033-00085, Hepatobio bank), and the tumor banks of CHU Bordeaux (BB-0033-00036) and CHU Henri Mondor for contributing to the tissue collection. This work was supported by Institut National du Cancer (INCa) with the International Cancer Genome Consortium (ICGC LICA-FR project) and NoFLIC projects (PAIR HCC, INCa and ARC), INSERM with the Cancer et Environnement (plan Cancer) and HETCOLI projects (Tumor Heterogeneity and Ecosystem program). The group is supported by the Ligue Nationale contre le Cancer (Equipe Labellisée), Labex OncoImmunology (investissement d'avenir), Coup d'Elan de la Fondation Bettencourt-Shueller, the SIRIC CARPEM and Fondation Mérieux. J.S. was supported by the ICE project (BPI France).

## Author contributions

Study concept and design: E.L., J.Sh., V.R., S.I., and J.Z.-R. Acquisition of data: V.R., G.C., J.-F.B., E.T., D.B., V.M., J.Se., P.B.-S., S.P., D.A., V.P., S.I., J.-F.D., and J.Z.-R. Analysis and interpretation of data: E.L., J.Sh., V.R., E.T., Q.B., V.M., J.Se., and J.Z.-R. Drafting the manuscript: E.L., J.Sh., and J.Z.-R. Critical revision of the manuscript: E.L., J.Sh., V.R., G.C., J.-F.B., E.T., Q.B., D.B., V.M., J.Se., P.B.-S., S.P., D.A., V.P., S.I., J.-F.D., and J.Z.-R. Statistical analysis: E.L. and J.Sh. Obtained funding: J.Z.-R.

## Additional information

**Competing interests:** The authors declare no competing financial interests.

