## [Peer Review File · Nature Communications]

Reviewers' comments:

Reviewer #1 Expert in HCC:

General comments

In the present manuscript Letouze et al have addressed the topic of the interaction of environmental exposures and endogenous cellular processes with the genomic modifications associated with driving tumorigenesis in liver cancer. The authors report that: (1) whole genome sequencing analysis of 44 new and 264 published liver cancers identified 10 mutational and 6 structural rearrangement signatures showing distinct relationships with environmental exposures, replication, transcription, and driver genes; (2) liver cancer-specific signature 16, associated with alcohol consumption, displayed a unique feature of transcription-coupled damage and was the main source of CTNNB1 mutations; (3) numerous indels were identified in very highly expressed hepato-specific genes, likely resulting from replication transcription collisions; (4) reconstruction of sub-clonal architecture revealed mutational signature evolution during tumor development exemplified by the vanishing of aflatoxin-B1 signature in African migrants; and (5) chromosome duplications were late clonal events and may represent rate limiting events in tumorigenesis. Based on these data the authors conclude that "These findings shed new light on the natural history of liver cancers". This conclusion is, at least in the broadest sense, true, but certainly not unexpected and/or surprising e.g. mutational signature evolution during tumor development associated with the vanishing of aflatoxin-B1 signature in African migrants. In light of the availabilities of extensive databases (including TCGA) on human HCC it would appear, at least to this review, that the observations reported in the current well executed work could be expanded and more rigorously validated by using these databases.

Reviewer #2 Expert in HCC and evolution:

In this manuscript, the authors performed whole genome analyses in newly sequenced 44 liver tumors and 264 published liver cancers, identified 10 mutational and 6 structural rearrangement signatures showing distinct relationships with environmental exposures, replication, transcription, and driver genes. They also demonstrated a unique feature of transcription-coupled damage in signature 16; a mutational signature evolution during tumor development; and regarded chromosome duplications as a rate limiting events in tumorigenesis. These findings are interesting, and helpful to understand the development of liver cancers from diverse etiologies. However, there are some major concerns:

1. In the present study, the authors only sequenced the genomes of 44 liver tumors, which included 35 HCC with diverse etiological backgrounds. Four fibrolamellar carcinomas (FLC) and 5 hepatocellular adenomas (HCA) were included for comparison. As a genomic study, this small sample size, even together with 264 published liver cancers, is not enough to support so many important conclusions.
2. In this manuscript, "most of 35 HCC developed in absence of cirrhosis (25 non-fibrotic, 7 chronic hepatitis and 3 cirrhotic livers) and associated with diverse etiological backgrounds including HBV (n=5) or HCV infection (n=4), alcohol (n=12), metabolic syndrome (n=7), hemochromatosis (n=1) and without etiology (n=6)". These etiological backgrounds are much different from the literatures, in which HBV or HCV infection is the major cause of HCC, and most of them develop in cirrhotic livers. What are the situations of the remaining "264 published liver cancers"?
3. Ten mutational signatures were identified in this manuscript. Besides with the etiological backgrounds, any possible association of them with clinicopathological features and classical prognostic indicators? And, the association between Signature 16 and CTNNB1 mutations has been reported in their previous literature (Schulze K et al. Nat Genet. 2015, 47(5): 505–511).

4. In Fig. 4, the authors concluded that functional selection of Beta-catenin activation may occur at different steps of hepatocarcinogenesis from the analysis of very small samples.

5. In Fig. 6, the authors showed that synchronous gains of chromosome duplications were late events during tumor progress, and were associated with bigger tumors (Fig. 6c). However, bigger tumor size does not necessarily exactly correlated with later stage in tumor progress. The analysis of correlations between synchronous gains of chromosome duplications with pathological staging of HCC may be better.

Reviewers' comments:

Reviewer #1:

General comments

In the present manuscript Letouze et al have addressed the topic of the interaction of environmental exposures and endogenous cellular processes with the genomic modifications associated with driving tumorigenesis in liver cancer. The authors report that: (1) whole genome sequencing analysis of 44 new and 264 published liver cancers identified 10 mutational and 6 structural rearrangement signatures showing distinct relationships with environmental exposures, replication, transcription, and driver genes; (2) liver cancer-specific signature 16, associated with alcohol consumption, displayed a unique feature of transcription-coupled damage and was the main source of CTNNB1 mutations; (3) numerous indels were identified in very highly expressed hepato-specific genes, likely resulting from replication transcription collisions; (4) reconstruction of sub-clonal architecture revealed mutational signature evolution during tumor development exemplified by the vanishing of aflatoxin-B1 signature in African migrants; and (5) chromosome duplications were late clonal events and may represent rate limiting events in tumorigenesis.

Based on these data the authors conclude that “These findings shed new light on the natural history of liver cancers”. This conclusion is, at least in the broadest sense, true, but certainly not unexpected and/or surprising e.g. mutational signature evolution during tumor development associated with the vanishing of aflatoxin-B1 signature in African migrants.

We fully agree with the Reviewer that it is not surprising to see a decrease of the activity of aflatoxin B1 signature in late mutations of African migrants tumors. What was unknown however is the magnitude of this change. Here we show that in two cases signature 24 is almost completely absent in subclonal mutations. Thus, subclonal diversification in these tumors is driven by completely different mutational processes as compared with the initial clonal development, which we believe is conceptually interesting.

In light of the availabilities of extensive databases (including TCGA) on human HCC it would appear, at least to this review, that the observations reported in the current well executed work could be expanded and more rigorously validated by using these databases.

We agree and we had included in the first version all the whole genome data sets that were publicly available. Another data set including whole genome HCC sequences from 49 Chinese patients exposed to aflatoxin B1 has been published recently (Zhang *et al.*, *Gastroenterology* **153**, 249–262.e2, 2017) but we have not yet received authorized access to these data despite our request on April 19th.

Of course the TCGA HCC data set, published in Cell on June 15th during the first

revision of this paper, is an amazing resource for the field. We have now taken advantage of this series to put together a validation series of 573 tumors analyzed by whole exome sequencing (WES), including 358 samples from the TCGA project and 215 tumors that we published earlier (Schulze et al., Nat Genet 2015). Although WES is less powerful than WGS to estimate precisely the contribution of signatures and to distinguish close signatures like signatures 5 and 16, we were able to validate several findings in this data set:

1) We validated the association of signature 16 with gender ($P = 3.2 \times 10^{-4}$) and alcohol consumption ($P = 4.6 \times 10^{-4}$). Unfortunately, tobacco-smoking status is not available in the TCGA series so we could not validate the higher amount of signatures 4 and 16 in smokers.

2) We estimated the contribution of mutational signatures to driver genes, and we confirmed the enrichment of signature 16 in *CTNNB1* mutations as compared with other mutations in *CTNNB1*-mutated tumors ($P = 0.059$) and other mutations in *CTNNB1*-wild-type tumors ($P = 8.6 \times 10^{-7}$).

3) We validated the high indel rate in very highly expressed genes (FPKM > 100), and the enrichment of deletions of 2-5 bases as compared with indels occurring in other genes.

These new findings were added in the main text and in Supplementary Fig. 4 and Supplementary Fig. 6:

“In this line, we validated the association of signature 16 with male gender, alcohol and *CTNNB1* mutations in an independent whole exome sequencing (WES) cohort of 573 tumors, comprising TCGA data²² and our previously published series¹¹ (**Supplementary Fig. 4**).”, page 4.

“We validated the high amount of indels, in particular deletions of 2-5 bases, in very highly expressed genes in an independent series of 573 tumors analyzed by WES (**Supplementary Fig. 6**).”, page 5.

Supplementary Figure 4: Validation of correlations between mutational signatures, risk factors and driver genes in an independent whole exome series. (a) Association between the number of mutations attributed to mutational signature 16 and gender. **(b)** Association between the number of mutations attributed to mutational signature 16 and alcohol consumption. **(c)** Distribution of mutational signatures associated with driver gene mutations. We estimated the probability of each driver gene mutation being due to each mutational process. We then summed these probabilities over all mutations and signatures to obtain the cumulative probabilities across all driver gene mutations (pie chart) and for each driver gene separately (barplot). **(d)** *CTNNB1* mutations (left) overall have higher probabilities being due to signature 16 than other mutations in the same samples (middle) and in other samples (right). The violin plots represent the distribution of probabilities for each group of mutations and horizontal segments highlight median values.

Supplementary Figure 6: Validation of the high amount of indels associated with very highly expressed genes in an independent whole exome series. (a) Number of single nucleotide variants (SNVs, left) and small insertions and deletions (indels, right) per megabase in genes as a function of expression level. Genes with expression between 0 and 100 fragments per kilobase of exons per million reads (FPKM) were divided in 5 gene expression quintiles. A separate group was created for very highly expressed genes (FPKM \geq 100). Error bars indicate the 95% confidence intervals of the estimated mutation rates. (b) Proportion of mutations being indels as a function of expression level. For each gene expression group, the proportion of mutations being indels was estimated as the number of indels divided by the number of SNVs + indels in genes belonging to the expression group. Error bars indicate the 95% confidence intervals of the estimated mutation rates. (c) Distribution of indel types and sizes in very highly expressed (FPKM \geq 100) versus all other genes.

Reviewer #2:

In this manuscript, the authors performed whole genome analyses in newly sequenced 44 liver tumors and 264 published liver cancers, identified 10 mutational and 6 structural rearrangement signatures showing distinct relationships with environmental exposures, replication, transcription, and driver genes. They also demonstrated a unique feature of transcription-coupled damage in signature 16; a mutational signature evolution during tumor development; and regarded chromosome duplications as a rate limiting events in tumorigenesis. These findings are interesting, and helpful to understand the development of liver cancers from diverse etiologies. However, there are some major concerns:

1. In the present study, the authors only sequenced the genomes of 44 liver tumors, which included 35 HCC with diverse etiological backgrounds. Four fibrolamellar carcinomas (FLC) and 5 hepatocellular adenomas (HCA) were included for comparison. As a genomic study, this small sample size, even together with 264 published liver cancers, is not enough to support so many important conclusions.

We fully agree with the Reviewer that the sample size for specific tumor types (FLC, HCA) and risk factors (e.g. hemochromatosis) is limited in our study. Thus, additional sequencing efforts will be necessary for these tumor groups and are likely to reveal new signatures and processes not yet identified with these series. The limitations are now clearly stated in the Discussion section:

“A limitation of this study is the small sample size for some etiologies. By grouping our new WGS data set with the ICGC-Japan series, we obtained substantial sample sizes for the main etiologies in France and Japan: alcohol, HBV and HCV infection. However, other etiologies like metabolic syndrome and hemochromatosis were poorly represented. Future whole genome studies will need to include more samples from other geographic areas associated with different risk factors, including nonalcoholic steatohepatitis that accounts for the rise of HCC cases in the US³⁹.”, page 9.

With respect to hepatocellular carcinomas, however, we have put together using ICGC data a substantial series of 299 tumor genomes, the largest series studied so far. With whole genome sequencing data, we had access to 3,964,651 somatic mutations overall and 26,074 structural rearrangements. This high number of events allowed us to identify highly significant associations between signatures, risk factors, genomic covariates and driver genes. To strengthen our analysis, we have now used as validation series a compendium of 573 tumors analyzed by whole exome sequencing, comprising TCGA data (358 tumors) and our series of 215 published HCC (Schulze *et al.*, Nat Genet 2015). In this independent series, we have been able to validate several key findings, including

the higher amount of signature 16 mutations in males and alcohol drinkers, the higher propensity of this process to generate *CTNNB1* mutations, and the specific indel-generating process affecting very highly expressed genes (see our response to Reviewer #1 for a more detailed description).

These validations have been added in the main text and in the new Supplementary Fig. 4 and 6.

2. In this manuscript, “most of 35 HCC developed in absence of cirrhosis (25 non-fibrotic, 7 chronic hepatitis and 3 cirrhotic livers) and associated with diverse etiological backgrounds including HBV (n=5) or HCV infection (n=4), alcohol (n=12), metabolic syndrome (n=7), hemochromatosis (n=1) and without etiology (n=6)”. These etiological backgrounds are much different from the literatures, in which HBV or HCV infection is the major cause of HCC, and most of them develop in cirrhotic livers. What are the situations of the remaining “264 published liver cancers”?

Indeed, we have voluntarily selected in our series etiologies that were less represented in available data sets to gain insights into the molecular diversity of HCC. The 264 published cases that we included were from a Japanese series (Fujimoto *et al.*, Nat Genet 2016) and are thus mostly associated with HBV (n=78, 30%) and HCV (n=149, 56%). Fibrosis stages according to the New Inuyama Classification were 12 F0 (5%), 31 F1 (12%), 66 F2 or F1-F2 (25%), 62 F3 (23%) and 93 F4 (cirrhosis, 35%).

In the revised manuscript, we have added a table describing the number of cases of each etiology in the two cohorts (Supplementary Table 5) and explained the specificities of the ICGC-Japan cohort in the main text:

“To correlate signature intensities with clinical and genomic features, we quantified the contribution of the 10 COSMIC signatures in a combined WGS data set comprising our 35 HCC and 264 HCC from the ICGC-Japan series (ICGC-JP), mostly related to HBV (30%) and HCV (56%) infection (Supplementary Table 5)¹⁹, page 3.

Sup Table 5: Comparison of clinical annotations between the INSERM and ICGC-Japan HCC series

Clinical feature		INSERM series (n= 35 HCC)	ICGC-Japan (n=264 HCC)	P-value	Test
Gender	Male	26 (74%)	198 (75%)	1	Fisher's exact test
	Female	9 (26%)	66 (25%)		
Age (median)		69	68	0.59	Wilcoxon rank-sum test
Alcohol	Yes	16 (46%)	105 (42%)	0.72	Fisher's exact test
	No	19 (54%)	145 (58%)		
HBV	Yes	4 (11%)	78 (30%)	0.026 (Fisher's exact test)	Fisher's exact test
	No	31 (89%)	186 (70%)		
HCV	Yes	5 (14%)	149 (56%)	2.59E-06	Fisher's exact test
	No	30 (86%)	115 (44%)		
Tobacco	Yes	15 (58%)	143 (57%)	1	Fisher's exact test
	No	11 (42%)	107 (43%)		
	Missing	9	14		
Fibrosis stage*	F0	0 (0%)	12 (5%)	1.15E-05	Chi-square test
	F1 or F0-F1	25 (71%)	31 (12%)		
	F2 or F1-F2	0 (0%)	66 (25%)		
	F3 or F2-F3	7 (20%)	62 (23%)		
	F4	3 (9%)	93 (35%)		
Tumor size (mm, median)		90	30	3.25E-11	Wilcoxon rank-sum test
Edmonson grade	I-II	11 (31%)	185 (70%)	1.39E-05	Fisher's exact test
	III-IV	24 (69%)	78 (30%)		
	Missing	0	1		
Vascular invasion	Yes	21 (60%)	89 (34%)	0.0047	Fisher's exact test
	No	14 (40%)	171 (66%)		
	Missing	4	0		

* According to METAVIR score for the INSERM series, and the New Inuyama Classification for the ICGC-Japan series.

3. Ten mutational signatures were identified in this manuscript. Besides with the etiological backgrounds, any possible association of them with clinicopathological features and classical prognostic indicators? And, the association between Signature 16 and CTNNB1 mutations has been reported in their previous literature (Schulze K et al. Nat Genet. 2015, 47(5): 505–511).

We thank the Reviewer for this interesting suggestion. We have now tested the association of molecular signatures with fibrosis stage, tumor size, Edmonson grade and vascular invasion. Satellite nodules and BCLC stage were not available for the ICGC-Japan series. We found a significant association between the amount of mutations attributed to signature 5 and Edmonson grade ($P = 0.0067$) and between signature 16 and tumor size ($P = 0.0066$). These new findings have been added as Supplementary Fig. 3 and described in the main text:

“Finally, correlations with clinicopathological feature revealed an increase of signature 5 with higher Edmonson grade (poorly differentiated tumors, $P=0.0067$) and a positive correlation between signature 16 and tumor size ($P=0.0066$, **Supplementary Fig. 3**).”, page 4.

Supplementary Figure 3: Correlation of mutational signatures with clinicopathological features. We assessed correlations between the amount of mutations attributed to each mutational signature and clinicopathological features (fibrosis stage, tumor size, Edmonson grade and vascular invasion). Only two significant associations were detected between signature 5 and Edmonson grade (**a**) and between signature 16 and tumor size (**b**).

Regarding the association of signature 16 with *CTNNB1*, we had indeed noted in Schulze's paper that *CTNNB1*-mutated tumors had a higher amount of mutations related to signature 16. However, this finding did not necessarily imply that *CTNNB1* mutations were directly caused by signature 16. It could have been an indirect correlation, e.g. signature 16 being due to alcohol and *CTNNB1* mutations being particularly advantageous in alcohol-exposed cells. In this work, whole genome sequencing data allowed us to quantify very precisely the proportion of each signature in each tumor, and to estimate the probability of each individual mutation being due to each signature. We were thus able to show that *CTNNB1* mutations themselves are more likely to be due to signature 16 than other coding mutations, even when restricting to *CTNNB1*-mutated tumors only (Fig. 1f). This finding provides the missing piece of evidence to understand the link between alcohol and *CTNNB1*, showing that the alcohol-related signature 16 has a higher propensity of generating mutations at *CTNNB1* hotspots.

4. In Fig. 4, the authors concluded that functional selection of Beta-catenin activation may occur at different steps of hepatocarcinogenesis from the analysis of very small samples.

We fully agree that, in this paper, we provide only few cases to establish the timing of *CTNNB1* mutations along hepatocarcinogenesis. However, these are representative examples confirming previous findings from our lab. In 2014, we analyzed 111 classical adenomas (HCA), 9 borderline HCA/HCC lesions and 6

HCC resulting from HCA malignant transformation, showing that *CTNNB1* mutation was an early event in HCA, followed by *TERT* promoter mutations upon carcinoma progression (Pilati *et al.*, Cancer Cell 2014). In another study (Nault *et al.*, Hepatology 2014), we analyzed mutation frequencies in 88 premalignant lesions from low-grade dysplastic nodules to small and progressed HCC, showing that *TERT* promoter mutations were the earliest events already frequent in premalignant lesions whereas mutations in other driver genes, including *CTNNB1*, only arose in fully developed tumors. Our clonality analysis nicely confirms these previous conclusions. We have rephrased this part in the main text to clarify this:

“We previously showed that *TERT* promoter mutations are early events occurring in premalignant HCC lesions whereas other driver mutations, including *CTNNB1*, occur in later stages^{25,26}. By contrast, *CTNNB1* is frequently mutated in benign adenomas at risk of malignant transformation requiring in most of cases *TERT* promoter mutations for carcinoma development^{27,28}. Here, we analyzed two cases of adenoma to carcinoma progression by WGS (Fig. 4c). One carcinoma displayed subclonal acquisition of *TERT* promoter, *ARID2* and *PTEN* mutations in 13% of tumor cells. The other carcinoma progressed from a subclone representing 14% of the adenoma sample, without new driver gene mutation but with acquired duplications of chromosome 7 and 17q. These results show that tumor heterogeneity is more important in benign liver tumors, with carcinoma progression originating from one subclone, and, in line with our previous findings, suggest that functional selection of β -catenin activation may occur at different steps of hepatocarcinogenesis.”, page 6.

5. In Fig. 6, the authors showed that synchronous gains of chromosome duplications were late events during tumor progress, and were associated with bigger tumors (Fig. 6c). However, bigger tumor size does not necessarily exactly correlated with later stage in tumor progress. The analysis of correlations between synchronous gains of chromosome duplications with pathological staging of HCC may be better.

We thank the Reviewer for this interesting suggestion. We have now tested the correlation between synchronous gains of chromosome duplications and Edmonson grade. We found a significant enrichment of multiple synchronous duplications or whole genome duplications in tumors of higher Edmonson grade ($P = 0.017$, Chi-square test for trend). We have now added this new finding in the manuscript:

“By contrast, synchronous gains observed in 14 HCC (89% pmt), and whole genome duplications found in 5 tetraploid HCC (95% pmt) were very late events associated with bigger tumors (median size 11 cm vs 6 cm in other tumors, $P=0.016$, Fig. 6c) of higher Edmonson grade ($P=0.017$), and may represent rate-limiting steps allowing the expansion of the tumor to a detectable mass.”, page 7.

REVIEWERS' COMMENTS:

Reviewer #1 (Remarks to the Author):

N/A

Reviewer #2 (Remarks to the Author):

The authors have made appropriate adjustments in this revised manuscript to address most of my previous concerns. I will suggest the manuscript to be accepted for publication.

REVIEWERS' COMMENTS:

Reviewer #1 (Remarks to the Author):

N/A

Reviewer #2 (Remarks to the Author):

The authors have made appropriate adjustments in this revised manuscript to address most of my previous concerns. I will suggest the manuscript to be accepted for publication.

We thank the reviewers for time and help in improving our manuscript.